# Selection of primary health care quality indicators in Europe: A Delphi study protocol

**Mariana Lobo**[1]*, **Andreia Pinto**[2,3], **Glória Conceição**[3], **Sara Escadas**[3], **Adriane Mesquita de Medeiros**[4], **Bruna Dias**[5], **Marta Sousa Pinto**[6], **Emília Pinto**[1,7], **André Ramalho**[1,8], **António Pereira**[1,9,10], **Manuel Gonçalves Pinho**[11,12], **Pedro Castro**[1], **Vera Pinheiro**[2,3,13], **Paulo Santos**[1], **João Vasco Santos**[1,14], **Alberto Freitas**[1]

1 CINTESIS@RISE, MEDCIDS, Faculty of Medicine, University of Porto, Porto, Portugal, 2 CINTESIS, Faculty of Medicine, University of Porto, Porto, Portugal, 3 MEDCIDS, Faculty of Medicine, University of Porto, Porto, Portugal, 4 Faculty of Medicine, Federal University of Minas Gerais, Belo Horizonte, Brazil, 5 Ribeirão Preto College of Nursing, University of São Paulo, Ribeirão Preto, Brazil, 6 CINTESIS@RISE, Faculty of Psychology and Education Sciences, University of Porto, Porto, Portugal, 7 Pain Unit, Maputo Central Hospital, Maputo, Mozambique, 8 Applied Health Research and Innovation Unit, CEJAM - Research and Studies Center "Dr João Amorim", São Paulo, Brazil, 9 Family Health Unit, Unidade de Saúde Familiar Prelada, ACES Porto Ocidental, Porto, Portugal, 10 PHC – Comissioning Department Northern Regional Administration of Health, Porto, Portugal, 11 CINTESIS@RISE, Department of Clinical Neurosciences and Mental Health, Faculty of Medicine, University of Porto, Porto, Portugal, 12 Department of Psychiatry and Mental Health, Unidade Local de Saúde do Tâmega e Sousa, Penafiel, Portugal, 13 Health Unit of Matosinhos, Matosinhos Local Health Unit, Matosinhos, Portugal, 14 ACES Grande Porto V – Porto Ocidental, ARS Norte, Porto, Portugal

* nanalobo@gmail.com, marianalobo@med.up.pt

## Abstract

### Objective

To describe a protocol to achieve consensus on valid and important indicators to assess primary health care (PHC) quality regarding all clinical contexts of PHC in European health systems.

### Study design

Qualitative study using the Delphi technique to gain consensus among European panels of experts comprising a heterogeneous professional background.

### Methods

Potential candidate indicators were extracted and translated according to a set of informative elements (*i.e.*, name, description, formula, unit of analysis, and sources). This list was then independently reviewed, and duplicates were removed totaling 1726 indicators. To guarantee a good response rate, indicators were distributed across 57 Delphi panels organized by clinical context. Each panel is a Delphi process, assessing between 23 to 33 indicators. Experts' opinions on the validity and importance of the extracted indicators will be obtained through two rounds of online questionnaires, using a 9-point Likert scale and free-text boxes. To prevent biased responses, participation will be anonymous to other participants and to the team administrating panels. Consensus will be considered if at least 70% of

**Data Availability Statement:** All relevant data are within the manuscript and its Supporting information files.

**Funding:** This article was supported by National Funds through FCT – Fundação para a Ciência e a Tecnologia , I.P., within Cintesis, R&D Unit (reference UIDB/4255/2020).

**Competing interests:** The authors have declared that no competing interests exist.

ratings ($\geq$7 assuming 10 participants) lie within the 7–9 range and less than 15% of ratings (<2 assuming 10 participants) are in the 1–3 range. Analysis of results will be streamlined and generalizable across panels using scripts.

## Conclusions

This protocol will contribute to improve the quality of PHC in Europe by achieving a consensual and concise list of PHC quality indicators retrieved from the scientific literature that fit current clinical guidelines and populations' needs in countries from the European region according to the World Health Organization.

## Introduction

Primary care plays a central role in providing healthcare, contributing to health promotion and prevention. It differs from other care settings by providing continuous care in an integrated manner focused on the whole person, according to each patient's unique needs, culture, values, "and preferences in a particular family and community environment. Being the patient's first contact with the health system, primary health care (PHC) practices coordinate care within the health system across other specialty care, hospitals, among others as deemed necessary by PHC providers [1–3]. Previous studies indicate that good quality PHC systems are associated with lower hospitalization rates, fewer emergency visits, lower total costs, improved satisfaction both from patient and healthcare providers, improved adherence to clinicians' recommendations, improved health and better self-reported health outcomes, greater equity, less service duplication, higher overall efficiency, improved patient self-management for chronic conditions, among others [2]. Due to its ability to maximize the quality of health care delivered and the well-being of populations, enhancing PHC is increasingly a policy priority for health systems around the globe [4].

Health indicators are an important tool for any healthcare system to improve performance and the quality of the healthcare provided to patients. These consist of measurable elements of the structures, processes, and outcomes of care and are usually operationalized as a ratio between a numerator and a denominator specifying the population at risk [5].This definition is in keeping with the Donabedian framework for quality improvement and with the definition of quality of care as "the degree to which health services increase the likelihood of desired results and are consistent with current professional knowledge," proposed by the National Academy of Medicine, former Institute of Medicine in 1999 [6].

Monitoring health indicators has become indispensable in health policy activities to raise awareness, foster transparency and accountability for the quality of healthcare delivered, and incentivize improving healthcare quality. Additionally, health policies defining financial incentives based on health quality indicators, such as pay-for-performance programs or penalties based on performance level targets, are becoming more common [7, 8]. As of 2018, thirteen OECD countries have introduced new payment models to encourage high-quality primary health care. However, this requires the right set of measures covering all health care provision and coordination aspects, including patient-reported experiences and outcomes measures [9].

There is a vast collection of studies in the scientific literature proposing health indicators to assess the quality of primary care. Ramalho *et al.* have recently performed an overview of systematic reviews assembling 727 indicators proposed to measure the quality of PHC [10]. Monitoring so many indicators is complex and time-consuming. Hence a shorter setlist of indicators

could facilitate this task and would be of great value to health policymakers, healthcare providers, and patients to inform about the quality of PHC. Moreover, indicators retrieved from the scientific literature may no longer fit the current clinical guidelines and populations' needs [11, 12]. Specifically, internationally developed indicators may not apply or have little relevance within the context of a health system of a specific country since health systems worldwide vary in their organization, financing systems, and population needs [13]. However, international comparisons are important instruments to public health and health policy, exposing areas of health care that need improvement and readily elucidating how to achieve better healthcare quality and health. Such comparisons became ubiquitous, especially due to efforts of international organizations such as the OECD Statistics (https://stats.oecd.org/), the WHO Observatory on Health Systems and Policies (https://www.euro.who.int/en/about-us/partners/observatory), the GBD project (http://www.healthdata.org/gbd), among others. Sound comparisons across countries require a common set of valid measures that are relevant to and feasible in those countries [14]. Particularly, countries that share global strategic synergies to improve the health of citizens, sharing many common features regarding PHC, such as the fact that it is either the filter for access to specialists or users have advantages in using it for that end, can greatly benefit from utilizing a standard list of PHC quality indicators.

## Study objectives

This study will use the Delphi technique to produce a validated selection of quality indicators to assess primary health care in countries from the European region (https://www.who.int/countries). Specifically, we will derive consensus among experts regarding the validity, including face and content validity, and importance of quality indicators previously identified through an overview of systematic reviews [10]. Based on the patterns of response across countries and experts' profiles, we will discuss the applicability of the selected indicators for within and between countries comparisons.

## Methods

### Preparation of candidate indicators

Instead of developing indicators for PHC, these have been abstracted from the literature by our research team. This process has considered previous work carried out in an umbrella review of systematic reviews covering PHC quality indicators and the primary studies of the reviews included in this umbrella [10]. In total, 1726 indicators (S1 Table) were identified, covering different types of indicators (*e.g.*, outcome, process, and structure), capturing different types (e.g., acute, chronic, preventive) and purposes (e.g., screening, diagnosis, treatment, follow up) of care, as well as, different domains of quality of care, and covering several clinical contexts (*i.e.*, WHO ICPC-2 chapters categorization) [15].

Our team examined and translated indicators extracted according to a set of informative elements (*i.e.*, name, description, formula, unit of analysis, and sources). The list of indicators was then reviewed by five independent medical doctors, who also removed any remaining duplicates. This process was carried out because we believe that enforcing this consistency among indicators would improve the interpretability and comparison of indicators by the Delphi participants.

### Study design

This protocol proposes the Delphi process to select PHC quality indicators [16], which will comprise 57 Delphi panels covering indicators of different clinical areas to be administered to experts in PHC (comprising different PHC stakeholders) working in European countries.

The Delphi process seeks to obtain consensus from the opinions of experts. Traditionally, the process consists of a series of structured questionnaires (also known as rounds), that each participant responds anonymously and iteratively. Anonymizing individual responses is essential to prevent bias, as participants may be influenced if the identity of other respondents is known. At the end of each round, the panel responses are summarized and shared with the participants, who are then allowed to respond in light of both the individual and the group's responses. The process ends when a consensus is achieved [17–19]. More recently, the real-time Delphi process has been proposed as a more efficient variation of the traditional Delphi process. Essentially, it allows participants to review their answers in a questionnaire considering the real-time group's response [20].

The Delphi study will be conducted online using a software to aid with the communication with participants, improving access, data collection and results sharing. Two rounds of questionnaires will be administered anonymously to participants via the eDelphi program (www.edelphi.org). The rounds will use 9-point Likert scales to evaluate the level of importance and validity of quality indicators, alongside free text boxes. The first round will be carried out as a traditional Delphi round to ensure anonymity and prevent biased ratings. The second round will be carried out in real-time to support an interactive and efficient process that simultaneously ensures effective discussions between participants.

All rounds will allow for asynchronous answering so that one participant can take part several times and change their answers until the end of a given time frame (expected time of two weeks) is reached. This time frame may be expanded if a low response rate is experienced.

## Definitions of the selection criteria

The selection criteria that participants will have to consider are importance and validity. Definitions of these criteria will be provided at the beginning of each round as follows:

**Importance**: The indicator should capture important performance aspects of primary health care. It should address areas of concern of policymakers where there is a clear gap and potential for improvement through identifiable events associated with the healthcare system. It should also permit useful comparisons within and across countries [21, 22].

**Validity**: The indicator is valid if it is clinically logical and supported by consensus (i.e., face validity); represents a valid measure supported by evidence demonstrating a correlation with the quality of care (i.e., content validity); is reproducible by and comparable across different countries, organizations, and providers, as well as using different data sources over time (i.e., reliable/consistent); detects important changes in the quality of care discriminating well different levels healthcare quality (i.e., sensitivity and specificity) [5, 23, 24].

## Development of the Delphi questionnaires/panels

The final list of candidate indicators has been ordered by clinical context and similarity and then distributed across fifty-seven Delphi questionnaires/panels. The number of panels per clinical context varied between 1 to 11 panels, depending on the number of existing indicators for each of the contexts, and each panel included between 23 and 33 indicators. Since previous studies using the Delphi technique to select healthcare quality indicators varied greatly in the number of indicators included in the first questionnaire [16], the number of indicators per panel was determined based on a compromise between the response/drop-out rate and the number of Delphi panels. The ideal number of indicators estimated to minimize dropouts was 25–30 indicators per panel. In most cases, all the indicators assigned to one panel correspond

to a single clinical context to make the evaluation more meaningful by comparison of context-wise indicators. However, some panels combined indicators from different clinical contexts because the number of indicators regarding some contexts was small (S1 File).

**eDelphi.** The eDelphi (eDelphi.org) is a subscription license online program specifically designed for the qualitative use of the Delphi method affording complete anonymous participation (responses are tagged with an anonymized id). It has plenty of options that allows to streamline the entire panel management of a Delphi process, which was a key consideration in this study, given the large number of panels to implement. Specifically, 1) managing invitations and reminders are simplified with the eDelphi; 2) the acceptance link that is sent to participants is personal, hence allowing them to respond to a questionnaire in the eDelphi without the need to register/login in the program; 3) exporting results as an image (bubble plot) or a google sheet. It is a convenient feature to feedback results with participants since plots contain descriptive measures of the group responses (Q1, Q2, Q3) over two dimensions; and 4) to analyze consensus on a regular basis with a statistical software of our choice.

Several limitations of the eDelphi have also been identified. First, since participation in a panel is anonymous, it will be virtually impossible to distinguish between participants that complete the questionnaire from those that respond to some questions (or even enter the questionnaire but do not respond to any question). Therefore, reminders will be sent to all participants who have accepted to participate in the round of a panel, unless participants disclose their identity (e.g., using the link to reject participation after having accepted or sending us an email). The only way we could overcome this limitation would be to export results regularly to link new answers with new participants. Second, because experts participate anonymously, the individual responses made in the first round can only be presented to the respective participant in the second round if the same type of question is chosen in round 2. Third, the Delphi does not provide a design mode to implement mandatory questions; hence participants can submit blank questionnaires. Since the participation is anonymous, we won't be able to ask participants to complete specific information. Participants with all questions blanks will be removed from the analysis at the end of each round.

## Study population

**Participants.** We will invite general practitioners and public health physicians, nurses, and other healthcare professionals, as well as researchers and other stakeholders in healthcare quality assessment living or working in European countries (S1 File). Panel members will be selected from different fields relating to clinical practice, health policy, and academia. We will recruit as many experts as required to achieve a 10-rating minimum per quality indicator in each Delphi panel. This is in line with the Rand UCLA criteria method that advocates that nine experts consist of a sufficiently large panel to achieve consensus and to incorporate the necessary heterogeneity needed to ensure study quality [25]. For a total of 57 Delphi panels, the expectation is that we will need to invite a minimum of 2300 participants (approximately 40 participants per panel) based on an acceptance rate of 30% (based on a related pilot study) and a drop-out rate of 10% [26]

**Recruitment.** To reach the targeted experts, we will use contacts in the public domain of members of European and National Professional Associations and related working networks and groups (*e.g.*, EUPHA, WONCA, Nurses associations), editors of journals classified as PHC according to the Journal Citation Report (https://jcr.clarivate.com/jcr/browse-category-list), as well as experts identified in PHC conferences. If a contact is not immediately available on the website of the organization considered, scientific articles published by target experts will be

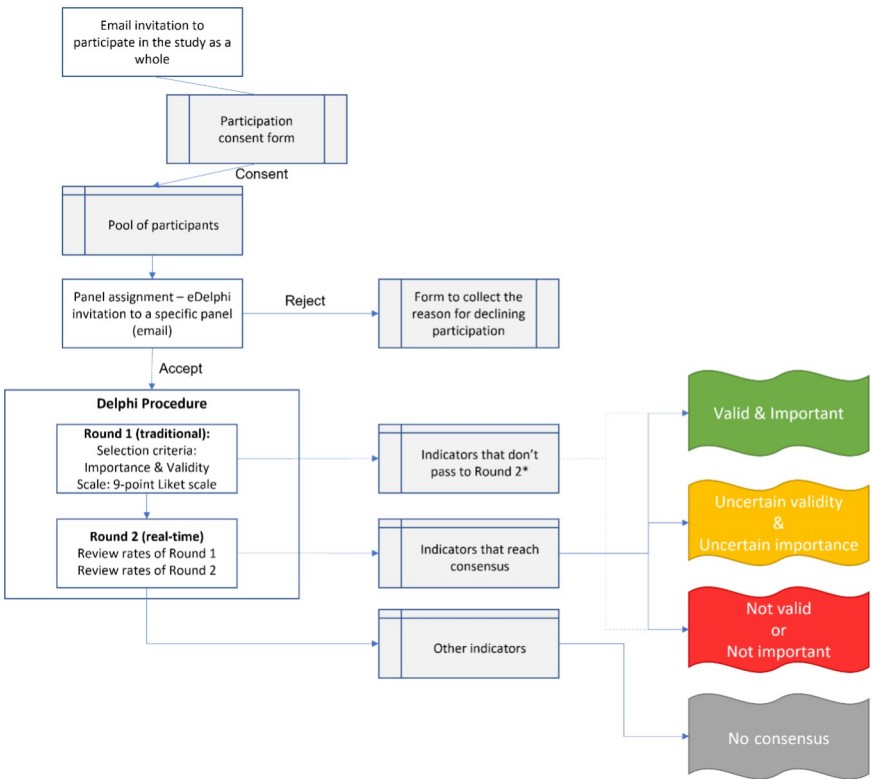

**Fig 1. Delphi study diagram flow to select PHC indicators.** *Refers to Decision matrix in Table 1.

searched using the name and affiliation information to identify a (potential) useable email contact.

Formal invitations to participate in the study will be sent by email (S1 File), which will include a link to a consent form containing information about this study and a brief sociodemographic questionnaire (S2 File). To increase sample size, we will also use a "exponential non-discriminative snowball sampling" approach by asking the expert contacted initially to forward the invitation email to their peers. Experts consenting to participate in the study will constitute the pool of potential participants to be distributed across the different Delphi panels. Participants completing a specific panel will be asked if they would like to participate in other panels. (Fig 1).

## Study procedure

Fig 1 summarizes the study procedure. We will use the eDelphi program to develop online questionnaires for each round of each panel and to invite potential participants from the pool of experts consenting to participate in the study. Experts will be invited to respond to a round of a panel via eDelphi by email with a link to accept and a link to reject participating in that round (S1 File). The email will include information about the study aim, the expected time for completion of the questionnaire, the ethics approval of the study, a request to complete each round within two weeks, and our commitment to share results at the end of the study. Reminders will be sent on a weekly basis. Participants rejecting or explicitly withdrawing (after accepting participating) from the panel will be emailed a link to a form requesting to provide a reason for declining participation (S3 File).

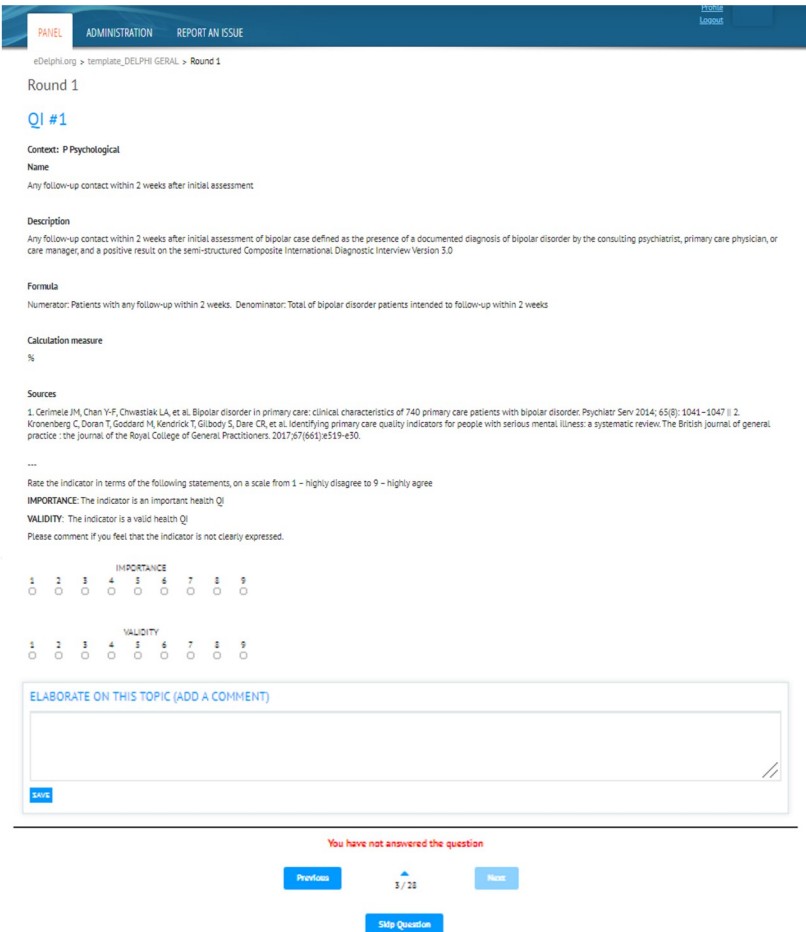

**Fig 2. Example of the eDelphi user interface for rating indicators in round 1.**

**Round 1.** The first round will be carried as a traditional Delphi round, where experts participate anonymously. An initial set of 40 invitations per panel will be sent assuming that this number will be enough to guarantee ten ratings per indicator. To promote a diverse and inclusive panel, invitations will be addressed to experts from at least two (ideally more) European countries and will comprise at least one general practitioner/family doctor, one public health doctor and one nurse.

In round 1, participants will initially be informed about the study and how to answer the questionnaire through a brief description of the structure of the questionnaire and selection criteria (S1 File), an information sheet (S4 File) and video tutorial (https://youtu.be/SaKSkyrYq9M) as well as they will be asked to provide demographic information including their gender, age, country of residence/work, level of education, professional area, and type of work institution. This will be necessary to assess experts' profiles in each round and panel.

Participants will then be asked to use radio buttons to rate indicators according to two criteria: 1) Importance and 2) Validity, expressing how much they agree with a statement defining each criterion on a scale from 1 –highly disagree to 9 –highly agree (Fig 2). Participants will be encouraged to leave comments in a free-text box to justify ratings between 1 and 3, if the indicators are not clearly expressed.

**Table 1. Decision matrix regarding indicators transitioning to round 2.**

| | | | VALIDITY | | | |
|---|---|---|---|---|---|---|
| | | | Consensus | | | No Consensus |
| | | | Not Valid | Uncertain | Valid | |
| IMPORTANCE | Consensus | Not Important | Remove | Remove | Remove | Remove |
| | | Uncertain | Remove | Remove | Keep | Keep |
| | | Important | Remove | Keep | Remove | Keep |
| | No Consensus | | Remove | Keep | Keep | Keep |

Participants who do not start answering the questionnaire (only login) while the round is open (after clicking the link to accept to participate in the round/panel) will not be invited to round 2.

An indicator will be removed from the second round depending on the number of criteria that reached consensus and the type of consensus reached, according to Table 1. Specifically, indicators that either reached consensus regarding the lack of "Validity" or "Importance" will be considered inadequate for assessing PHC quality and therefore will not pass to the second round. Additionally, indicators that in round 1 reach consensus in both "Validity" and "Importance", will be assumed appropriate to assess PHC quality, and for this reason will also not pass to round 2. Finally, indicators for which consensus regarding the uncertainty of both their "Validity" and "Importance" of indicators to assess PHC quality in round 1 will not pass to round 2 as well. Consensus in this case is more stringent (1 or none out of 10 participants rating outside the 4–6 range of values, see definition below). This is justified by the fact that we want an efficient process to identify indicators that are clearly good candidates to assess quality of PHC.

**Round 2.** Indicators with no consensus in either dimension will pass to round 2, except when consensus in the other dimension is either "Not Valid" or "Not Important". Indicators with consensus on importance but uncertain validity and vice-versa will also pass to round 2. (Table 1).

In round 2, the questionnaire will additionally include the results from round 1. To this end, we will use the eDelphi results export tool to plot aggregate responses for each indicator evaluated. The plots will be inserted in the corresponding question, along with anonymous quotes of all the comments received in round 1 (S1 File). One week will be needed to assemble the questionnaire for round 2. Provided that low response rates do not prove to be a problem, each second round will open three weeks after the first round.

The second round will be carried out in real-time. This means that the group response 2 will be updated in real-time and will be made visible to the participant immediately after giving their response. In practice, this allows experts to respond each question twice requiring a single interaction with the software (Fig 3) [20, 27].

In round two, participants will also initially have access to the same information about the study; the only difference will be that the information will include additional explanations about the real-time approach Delphi. Participants will also have to provide the same demographic information as in round 1. This will allow us to invite supplementary experts in round 2 if we experience dropouts.

Participants will then be asked to rate their level of agreement with the clinical indicators that have been passed on to round 2 regarding their validity and importance. The rating should consider the distribution of participants' ratings and comments from the previous round which will be visible in the description of each indicator (S1 File). To rate the indicators,

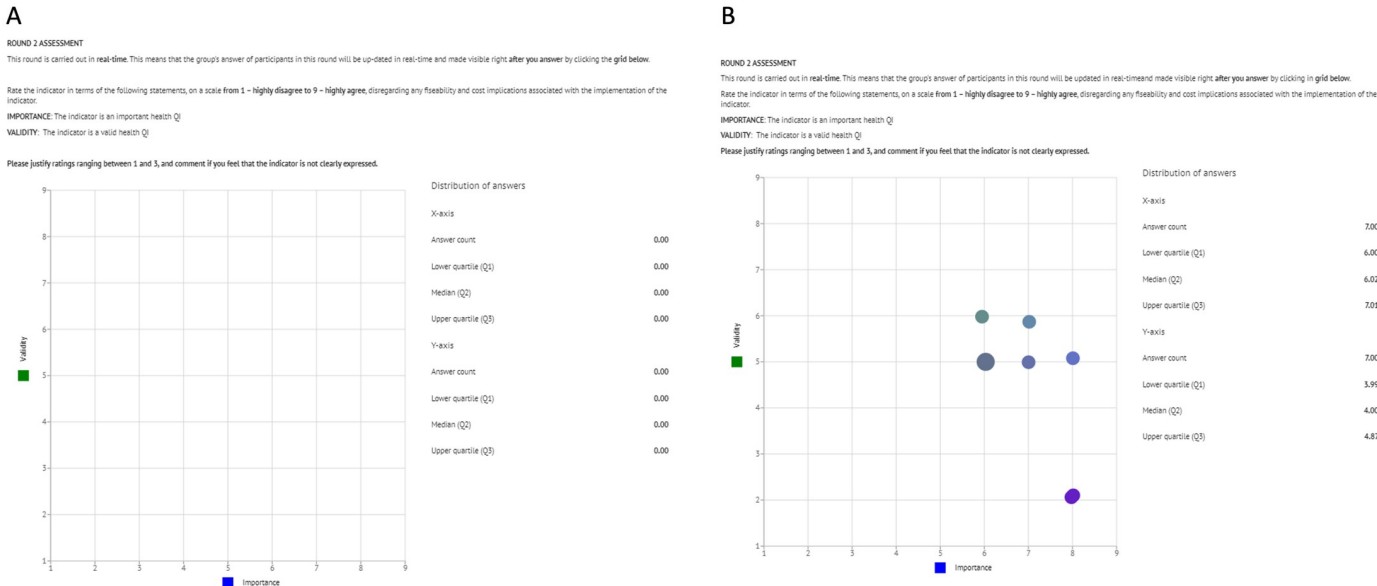

**Fig 3. Detail of the real-time Delphi user interface for rating indicators in round 2.** A) before start answering/clicking on the grid for the first time; B) right after answering/clicking on the grid for the first time, the group's response becomes visible, and the participant has the chance to review the initial response.

participants must point and click in a nine-by-nine grid representing the 2-dimension space of possible rating combinations in round 2. Ratings will be rounded to the nearest integer number (and this will be informed to participates at the beginning of round 2). After clicking/responding, de-identified ratings of other participants will appear on the grid. (Fig 3), which may elicit a change in the response.

## Data analysis plan

De-identified ratings will be summarized using eDelphi tools into quartiles 1, 2 and 3 and as the number of ratings in each indicator, which will then be reported to panel members as feedback after round 1 of each panel.

Consensus targets for this study are defined 'a priori' based on other studies [16, 25]. Consensus for each criterion will be determined based on the distribution of participants' ratings and the level of disagreement among the participants according to three categories.

- Indicators with 70% or more ratings in the 7–9 range and less than 15% of ratings in the 1–3 range will be classified as "Important" or"Valid".

- Indicators with 70% or more ratings in the 1–3 range and less than 15% of ratings in the 7–9 range will be classified as "Not important" or"Not valid".

- Indicators with 85% or more ratings in the 4–6 range will be classified as "Uncertain".

A minimum of 10 answers per indicator will be required to close a round unless consensus is reached with fewer answers and two weeks pass since starting the round. (Table 2).

Descriptive statistics (min, Q1, Q2, Q3, max) of the distribution of the number of responses, number of indicators that either have at least 10 answers or reached consensus, time a round is open, number of drop-outs, number of invitations sent will be monitored regularly to determine the progress of a round. This information may be used in the reminders to motivate

**Table 2. Consensus with fewer than 10 answers—Combination of number of responses across the three rating ranges.**

| Number of responses per range of rating | | | Total respondents | Consensus |
|---|---|---|---|---|
| 1–3 | 4–6 | 7–9 | | |
| 0 | 0 | 9 | 9 | Valid or Important |
| 0 | 1 | 8 | 9 | |
| 0 | 2 | 7 | 9 | |
| 0 | 9 | 0 | 9 | Uncertain validity or Uncertain Importance |
| 9 | 0 | 0 | 9 | Not valid or Not important |
| 8 | 1 | 0 | 9 | |
| 7 | 2 | 0 | 9 | |

experts to answer and close a round. Descriptive statistics of participants' profiles will be determined to check heterogeneity of participants' profiles within and between panels.

Any free text responses provided by participants to specific indicators during the two rounds will be analyzed thematically [28]. The thematic analysis of qualitative open responses will not be shared with participants, it will only be used to discuss the final results of panels for dissemination purposes of this work.

R Studio will be used to develop R scripts to implement repetitive and automatized analyses. Scripts will be distributed through the team for consistency of analysis across panels.

## Steering committee/project governance

A Steering Committee at the Faculty of Medicine, University of Porto, was formed to develop and conduct this project. The Committee includes representatives of various disciplines (Medicine, Nursing, and Data analysis) and research expertise (qualitative, quantitative, and Delphi methods). The research methods were established in face-to-face meetings and email communications. Different members of the committee were responsible for: A) extracting and processing indicators available in the literature, B) reviewing indicators definitions and removing indicator duplicates, C) identifying (contacts of) potential participants, recruitment, and developing the questionnaires, and D) defining a framework and choosing tools to manage Delphi panels. Several pilot tests were carried out to establish the organization, structure, and amount of information provided in questionnaires.

## Ethics and data management

This project has been approved by the ethics committee of the São João Hospital University Center/Faculty of Medicine, University of Porto [Ref. 338–19].

Personal data (names, email addresses, profession, working institution) will only be collected to approach potential participants, send out links to the questionnaire, and send out reminders. These data will be publicly available and as such will be stored unencrypted. However, all responses will be de-identified through the eDelphi and the team will not have access to the key to re-identify participants.

## Discussion

This protocol describes the research design for a Delphi process to obtain consensus from experts on a core set of valid and important health indicators to assess PHC quality selected from a list of indicators abstracted from the scientific literature. This protocol will add to the

current instruments aimed at incentivizing PHC quality. Specifically, the implementation of this protocol will arrive to a concise list of health quality indicators validated for European countries, and as such it will be essential for comparisons of PHC quality across countries [9]. Therefore, this research will contribute to overcoming the issue regarding the vast collection of indicators to assess PHC quality [10], identifying indicators retrieved from the scientific literature that may no longer fit the current clinical guidelines and populations' needs in European countries.

The Delphi technique is well suited for our purpose since it is commonly used in health research, particularly to select health indicators [16]. Unlike methods that rely on face-to-face meetings to form a group consensus, such as the nominal group technique [21, 29] and the UCLA RAND method [25, 30], the Delphi is a soloistic, anonymous process avoiding direct confrontation of the experts. Thus, participation is less susceptible to dominant opinions, and controlled feedback allows for gradually forming a considered opinion [18, 31]. Nevertheless, face-to-face discussions could be important to allow the debate of potentially different viewpoints, but they are unlikely feasible in this study due to the large list of indicators to assess. It should be noted, however, that in Round 2, experts will interact as they will have access to the results of other participants in real time and can change their classification accordingly.

Moreover, we believe that the use of an online program specially developed to carry out Delphi studies, such as the eDelphi, will be key to manage a large amount of Delphi panels consistently over time, offering easy participation to experts, and achieving a reasonable response rate. Panel members will be drawn from a variety of backgrounds ensuring a multidisciplinary perspective and validity across European countries [25]. However, the lack of clear guidelines about implementing Delphi remains objectionable [31, 32], publishing this protocol should strengthen the research design of this study.

This study will disregard any feasibility and cost implications associated with the implementation of the selected indicators. This study will also disregard potential valid and/or important indicators that are not described in the literature. Additionally, limited time and resources available preclude at present to carry a final Delphi process to further select from the final consensual valid and important indicators that will be derived in this study. However, the present work will allow us to discern between the best and worst candidate indicators described in the literature within a specific clinical context. Future work will be needed to address these limitations.

This paper describes the design of a European Delphi study to derive a consensual set of quality indicators of PHC. The impact of this study will ultimately aid standardization of PHC and benefit the quality of care delivered to patients across European countries.

## Supporting information

**S1 Table. List of indicators for the Delphi.**
(XLSX)

**S1 File. Delphi_protocolo_suppl.**
(DOCX)

**S2 File. Delphi study consent—Google forms.**
(PDF)

**S3 File. Reasons for declining participation—Google forms.**
(PDF)

**S4 File. Information sheet.**
(PDF)

## Author Contributions

**Conceptualization:** Mariana Lobo, Paulo Santos, João Vasco Santos, Alberto Freitas.

**Data curation:** Mariana Lobo, Andreia Pinto, Bruna Dias, Emília Pinto, André Ramalho, António Pereira, Manuel Gonçalves Pinho, Pedro Castro, Vera Pinheiro.

**Methodology:** Mariana Lobo, Andreia Pinto, Glória Conceição, Marta Sousa Pinto, Emília Pinto, André Ramalho, Manuel Gonçalves Pinho, Paulo Santos, João Vasco Santos, Alberto Freitas.

**Project administration:** Mariana Lobo, Glória Conceição.

**Resources:** Sara Escadas, Adriane Mesquita de Medeiros, Bruna Dias.

**Supervision:** João Vasco Santos, Alberto Freitas.

**Writing – original draft:** Mariana Lobo.

**Writing – review & editing:** Mariana Lobo, Andreia Pinto, Glória Conceição, Sara Escadas, Adriane Mesquita de Medeiros, Bruna Dias, Marta Sousa Pinto, Emília Pinto, André Ramalho, António Pereira, Manuel Gonçalves Pinho, Pedro Castro, Vera Pinheiro, Paulo Santos, João Vasco Santos, Alberto Freitas.

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
