## [Decision Letter · Decision Letter 0]

25 May 2024

PONE-D-23-33217Selection of primary health care quality indicators in Europe: a Delphi study protocolPLOS ONE

Dear Dr. Lobo,

Thank you for submitting your manuscript to PLOS ONE. After careful consideration, we feel that it has merit but does not fully meet PLOS ONE’s publication criteria as it currently stands. Therefore, we invite you to submit a revised version of the manuscript that addresses the points raised during the review process.

Please, address all suggestions made by the reviewers below.

We look forward to receiving your revised manuscript.

Kind regards,

Vivian Aline Mininel, PhD

Academic Editor

PLOS ONE

Journal Requirements:

Reviewers' comments:

Reviewer's Responses to Questions

**Comments to the Author**

1. Does the manuscript provide a valid rationale for the proposed study, with clearly identified and justified research questions?

Reviewer #1: Yes

Reviewer #2: Yes

2. Is the protocol technically sound and planned in a manner that will lead to a meaningful outcome and allow testing the stated hypotheses?

Reviewer #1: Yes

Reviewer #2: Yes

3. Is the methodology feasible and described in sufficient detail to allow the work to be replicable?

Reviewer #1: Yes

Reviewer #2: Yes

4. Have the authors described where all data underlying the findings will be made available when the study is complete?

Reviewer #1: No

Reviewer #2: Yes

5. Is the manuscript presented in an intelligible fashion and written in standard English?

Reviewer #1: Yes

Reviewer #2: Yes

6. Review Comments to the Author

You may also provide optional suggestions and comments to authors that they might find helpful in planning their study.

Reviewer #1: This is a generally well written protocol for a Delphi approach to consensus on primary care quality indicators. I have the following generally minor comments for the authors to consider addressing:

Abstract:

Methods section final sentence: Not clear what 57 Delphi panels are referred to here. You do explain later in the text but the reference in the abstract is vague. It is essential to explain here that it is not only 1 Delphi panel you are proposing and why. 57 is quite a big number and it may be worth it to consider reducing by merging specialities.

Second paragraph of the methods: ‘anonymous’ whom? Research team or participants or public (upon publishing)?

Conclusions: When referring to ‘Europe’ here please be more specific using what boundaries/country groupings.

Study objectives:

Briefly define ‘validated’ here even though you do define a bit later.

The paragraph that follows starting with ‘These countries share…’ move up in the introduction section before the objectives as this is part of the rationale for the study.

Study design:

1st paragraph: be more specific with ‘several’ Delphi panels

Reference to ‘administered anonymously’ needs explaining as above. What is the justification for anonymisation?

The order of the methods is a little confusing. My preference is to start with where the baseline indicators come from.

Under the detailed definition of ‘validity’: how do you propose to establish that participants will apply those criteria in their answers?

The first paragraph under ‘round 2’ under ‘study procedure’: would be good to phrase this more clearly – e.g. indicators reaching consensus either way will not pass to round 2.

‘Expected results’ – no need for this section? Feels like repetition

Reviewer #2: Study on a relevant theme, in a pertinent methodological approach. The authors stated that the project received no funding and no conflicts of interest. The wording is clear, objective, meets the cultured norms of the language and guidelines recommended for scientific reports. From a list of indicators abstracted from the scientific literature, the authors developed an original, unpublished and unpublished protocol in any other scientific medium that describes, in detail, the Delphi research design to obtain expert consensus on a set of valid and important health indicators to assess the quality of Primary Health Care in Europe. The results will allow us to indicate a concise list of validated health quality indicators for European countries and, as such, will be essential for comparisons of PHC quality between countries.

Regarding the method, the protocol describes, with high technical standards and rich details, the steps to be followed, including aspects related to the selection criteria of importance and validity that the participating indicators should contemplate; composition of the steering committee and governance of the project with participants from various disciplines (Medicine, Nursing and Data Analysis) and research experts (qualitative, quantitative and Delphi methods); development of the Delphi questionnaires/panels, with the elaboration of candidate indicators including extensive review of systematic reviews covering quality indicators of PHC and primary studies; use of the eDelphi online program; study population, inclusion and recruitment criteria; study procedure synthesized in Figure 1 with the flow of the Delphi study diagram to select PHC indicators; criteria for round 1; Decision matrix regarding the indicators in transition to round 2; round 2; Data analysis plan, consensus targets, statistical procedures.

The project was approved by the ethics and data management committee. It is recommended that authors explain how the data underlying the results of their research will be made available, without restrictions, as well as to state how they plan to share data from their study when it is completed or published.

7. PLOS authors have the option to publish the peer review history of their article (what does this mean?). If published, this will include your full peer review and any attached files.

Reviewer #1: No

Reviewer #2: No

---

## [Author Response · Author response to Decision Letter 0]

4 Jul 2024

PLOS ONE

July 4th, 2024

ID. No.: PONE-D-23-33217

Dear Dr. Vivian Aline Mininel

We are very grateful to you and the reviewers for your review of our manuscript. Your feedback is highly valued, and we are grateful for the time and expertise you dedicated to evaluating our work.

Please find below (purple) a point-by-point response to all the comments.

Academic Editor PLOS ONE 

Journal Requirements:

3. Please include captions for your Supporting Information files at the end of your manuscript, and update any in- text citations to match accordingly. Please see our Supporting Information guidelines for more information: http://journals.plos.org/plosone/s/supporting-information. 

We have reviewed and formatted the document according to PLOS ONE guidelines.

Reviewers' comments:

Reviewer's Responses to Questions

Comments to the Author 

1. Does the manuscript provide a valid rationale for the proposed study, with clearly identified and justified research questions? 

Reviewer #1: Yes 

Reviewer #2: Yes 

2. Is the protocol technically sound and planned in a manner that will lead to a meaningful outcome and allow testing the stated hypotheses? 

Reviewer #1: Yes 

Reviewer #2: Yes 

3. Is the methodology feasible and described in sufficient detail to allow the work to be replicable? 

Reviewer #1: Yes 

Reviewer #2: Yes 

4. Have the authors described where all data underlying the findings will be made available when the study is complete? 

Reviewer #1: No 

Reviewer #2: Yes 

We have made available all materials used for this protocol, even the candidate indicators are available as supplementary material in S1 file.

Any findings that result from the implementation of this protocol will be potentially published as a scientific article. However, we do not yet have results to share.

5. Is the manuscript presented in an intelligible fashion and written in standard English? 

Reviewer #1: Yes 

Reviewer #2: Yes 

6. Review Comments to the Author 

You may also provide optional suggestions and comments to authors that they might find helpful in planning their study. 

Reviewer #1

Reviewer #1: This is a generally well written protocol for a Delphi approach to consensus on primary care quality indicators. I have the following generally minor comments for the authors to consider addressing:

Abstract:

Methods section final sentence: Not clear what 57 Delphi panels are referred to here. You do explain later in the text but the reference in the abstract is vague. It is essential to explain here that it is not only 1 Delphi panel you are proposing and why. 57 is quite a big number and it may be worth it to consider reducing by merging specialities. 

Second paragraph of the methods: ‘anonymous’ whom? Research team or participants or public (upon publishing)?

Conclusions: When referring to ‘Europe’ here please be more specific using what boundaries/country groupings. 

Thank you for your suggestions, we have reviewed the abstract accordingly. The following sentences address the three comments to the abstract.

“To guarantee a good response rate, indicators were distributed across 57 Delphi panels organized by clinical context. Each panel is a Delphi process, assessing between 23 to 33 indicators.”

“To prevent biased responses, participation will be anonymous to other participants and to the team administrating panels.”

“This protocol will contribute to improve the quality of PHC in Europe by achieving a consensual and concise list of PHC quality indicators retrieved from the scientific literature that fit current clinical guidelines and populations’ needs in European countries from the European region according to the World Health Organization.”

Study objectives: 

Briefly define ‘validated’ here even though you do define a bit later.

The paragraph that follows starting with ‘These countries share...’ move up in the introduction section before the objectives as this is part of the rationale for the study. 

Thank you! The manuscript was revised accordingly.

The word validated in this section refers to the study design – Delphi method – which confers a validated subset list of indicators, that is consensual to experts participating in the study. Not sure if the reviewer was referring to “validity” mentioned ahead of “validated”. We have briefly defined it as suggested.

“Specifically, we will derive consensus among experts regarding the validity, including face and content validity, and importance of quality indicators previously identified through an overview of systematic reviews.”

Study design:

1st paragraph: be more specific with ‘several’ Delphi panels

Reference to ‘administered anonymously’ needs explaining as above. What is the justification for anonymisation? The order of the methods is a little confusing. My preference is to start with where the baseline indicators come from.

Under the detailed definition of ‘validity’: how do you propose to establish that participants will apply those criteria in their answers?

The first paragraph under ‘round 2’ under ‘study procedure’: would be good to phrase this more clearly – e.g. indicators reaching consensus either way will not pass to round 2.

‘Expected results’ – no need for this section? Feels like repetition 

Thank you, we have revised the manuscript accordingly to the reviewer’s suggestions, adding clarifications in the main text and moving subsections within the methods section. Specifically,

-with regards to the anonymization, we have clarified to whom participation is anonymous to (other participants and the team conducting the Delphi process).

- we propose that participants will apply the criteria by providing them with the definitions of criteria as described in the description of the study (at the beginning of each round), in the information sheet, and in the video tutorial, as well as by asking two questions for each indicator separately as explained in the manuscript

“Participants will then be asked to use radio buttons to rate indicators according to two criteria: 1) Importance and 2) Validity, expressing how much they agree with a statement defining each criterion on a scale from 1 – highly disagree to 9 – highly agree (Fig 2).”

To improve the manuscript, we now refer in the main text that the criteria definitions are available to the participants at the beginning of each round. A Figure of how this is provided to participants has been included as supplementary material in S2 File, S2.5 Fig.

“In round 1, participants will initially be informed about the study and how to answer the questionnaire through a brief description of the structure of the questionnaire and selection criteria (S2 File), an information sheet (S5 File, supplementary material) and video tutorial (https://youtu.be/SaKSkyrYq9M) as well as they will be asked to provide demographic information including their gender, age, country of residence/work, level of education, professional area, and type of work institution.”

- Regarding the first paragraph under ‘round 2’, we have rephrased the text to clarify that indicators moving to round 2 are determined according to Table 1. Some text was moved to the previous subsection. However notice that the decision regarding whether to move or not an indicator to round 2 is not as trivial as suggested by the reviewer.

“Indicators with no consensus in either dimension will pass to round 2, except when consensus in the other dimension is either “Not Valid” or “Not Important”. Indicators with consensus on importance but uncertain validity and vice-versa will also pass to round 2. (Table 1)

- Expected results were removed.

Reviewer #2: Study on a relevant theme, in a pertinent methodological approach. The authors stated that the project received no funding and no conflicts of interest. The wording is clear, objective, meets the cultured norms of the language and guidelines recommended for scientific reports. From a list of indicators abstracted from the scientific literature, the authors developed an original, unpublished and unpublished protocol in any other scientific medium that describes, in detail, the Delphi research design to obtain expert consensus on a set of valid and important health indicators to assess the quality of Primary Health Care in Europe. The results will allow us to indicate a concise list of validated health quality indicators for European countries and, as such, will be essential for comparisons of PHC quality between countries.

Regarding the method, the protocol describes, with high technical standards and rich details, the steps to be followed, including aspects related to the selection criteria of importance and validity that the participating indicators should contemplate; composition of the steering committee and governance of the project with participants from various disciplines (Medicine, Nursing and Data Analysis) and research experts (qualitative, quantitative and Delphi methods); development of the Delphi questionnaires/panels, with the elaboration of candidate indicators including extensive review of systematic reviews covering quality indicators of PHC and primary studies; use of the eDelphi online program; study population, inclusion and recruitment criteria; study procedure synthesized in Figure 1 with the flow of the Delphi study diagram to select PHC indicators; criteria for round 1; Decision matrix regarding the indicators in transition to round 2; round 2; Data analysis plan, consensus targets, statistical procedures.

The project was approved by the ethics and data management committee. It is recommended that authors explain how the data underlying the results of their research will be made available, without restrictions, as well as to state how they plan to share data from their study when it is completed or published.

---

## [Editor Report · Decision Letter 1]

13 Aug 2024

Selection of primary health care quality indicators in Europe: a Delphi study protocol

PONE-D-23-33217R1

Dear Dr. Lobo,

We’re pleased to inform you that your manuscript has been judged scientifically suitable for publication and will be formally accepted for publication once it meets all outstanding technical requirements.

Kind regards,

Vivian Aline Mininel, PhD

Academic Editor

PLOS ONE

---

## [Editor Report · Acceptance letter]

20 Aug 2024

PONE-D-23-33217R1 

PLOS ONE

Dear Dr. Lobo, 

I'm pleased to inform you that your manuscript has been deemed suitable for publication in PLOS ONE. Congratulations! Your manuscript is now being handed over to our production team.

Kind regards, 

on behalf of

Dr. Vivian Aline Mininel 

Academic Editor

PLOS ONE